# Review of Acoustic Emission Detection Technology for Valve Internal Leakage: Mechanisms, Methods, Challenges, and Application Prospects

**DOI:** 10.3390/s25144487

**Published:** 2025-07-18

**Authors:** Dongjie Zheng, Xing Wang, Lingling Yang, Yunqi Li, Hui Xia, Haochuan Zhang, Xiaomei Xiang

**Affiliations:** 1School of Mechanical Engineering, Sichuan University, Chengdu 610065, China; 19382214730@163.com (D.Z.); m17208296681@163.com (X.W.); 19188156926@163.com (L.Y.); 17711356313@163.com (H.Z.); xxm@scu.edu.cn (X.X.); 2School of Advanced Technology (SAT), Xi’an Jiaotong-Liverpool University (XJTLU), Suzhou 215000, China; yunqi.li2202@student.xjtlu.edu.cn

**Keywords:** valve internal leakage, acoustic emission detection technology, wavelet analysis, intelligent algorithms, gaussian process regression, valve condition monitoring

## Abstract

Internal leakage within the valve body constitutes a severe potential safety hazard in industrial fluid control systems, attributable to its high concealment and the resultant difficulty in detection via conventional methodologies. Acoustic emission (AE) technology, functioning as an efficient non-destructive testing approach, is capable of capturing the transient stress waves induced by leakage, thereby furnishing an effective means for the real-time monitoring and quantitative assessment of internal leakage within the valve body. This paper conducts a systematic review of the theoretical foundations, signal-processing methodologies, and the latest research advancements related to the technology for detecting internal leakage in the valve body based on acoustic emission. Firstly, grounded in Lechlier’s acoustic analogy theory, the generation mechanism of acoustic emission signals arising from valve body leakage is elucidated. Secondly, a detailed analysis is conducted on diverse signal processing techniques and their corresponding optimization strategies, encompassing parameter analysis, time–frequency analysis, nonlinear dynamics methods, and intelligent algorithms. Moreover, this paper recapitulates the current challenges encountered by this technology and delineates future research orientations, such as the fusion of multi-modal sensors, the deployment of lightweight deep learning models, and integration with the Internet of Things. This study provides a systematic reference for the engineering application and theoretical development of the acoustic emission-based technology for detecting internal leakage in valves.

## 1. Introduction

As a core component of the national energy security strategic system, the sustained and stable supply of oil and gas resources plays a fundamental supporting role in the high-quality development of the national economy. Taking China as an example, according to statistics from the National Energy Administration of China, the country’s crude oil output reached 204 million tons, and natural gas production amounted to 20.36 billion cubic meters in 2022 [1]. Currently, there exists a massive global refining and petrochemical industry chain, where fluid control equipment centered around industrial valves has become a critical module for ensuring the safety of energy transportation systems.

In recent years, frequent valve failure incidents in industrial production have led to severe consequences.

A tragic example occurred on 3 December 1984, at the Union Carbide pesticide plant in Bhopal, India. During maintenance work, personnel flushed the methyl isocyanate (MIC) storage tank pipeline filter without installing mandatory blanking plates, relying solely on valve closure. Due to internal valve leakage caused by corrosion, water entered the MIC storage tank and triggered an exothermic reaction with MIC. This led to rapid temperature and pressure increases, ultimately causing the rupture disk to burst and safety valves to open. The catastrophic MIC gas release resulted in 6495 deaths and 125,000 poisonings. Similarly, in 2020, a Louisiana chemical plant in the United States experienced a chlorine leak due to valve failure, causing significant environmental damage. These incidents demonstrate how valve failures can trigger major safety disasters, posing serious threats to both industrial safety and public welfare. Most recently, on 13 April 2024, Huayou Chemical Co., Ltd. in Inner Mongolia encountered a steam valve leakage during the commissioning of a new piperonyl ring production unit. The leaking steam caused over-temperature and over-pressure conditions in a dichloromethane-containing reactor, leading to chemical leakage and a subsequent fire that injured one worker.

Valve leakage failure refers to the loss of a valve’s inherent ability to block fluid/gas flow due to the deterioration of its original sealing system caused by wear, corrosion, or aging under high temperature/pressure conditions [2]. Valve leakage can generally be classified into two types: external leakage and internal leakage. External valve leakage refers to the escape of medium from the exterior of the valve due to failure of stem seals or valve body connections. This type of leakage predominantly occurs when the valve body undergoes aging or operates under overload conditions [3]. External valve leakage can be easily detected since the escaping medium is directly observable. However, internal leakage presents significant detection challenges, often remaining unnoticed while creating substantial safety hazards [3]. To detect internal leakage and prevent industrial accidents, valves must currently be removed from pipelines for periodic inspection, a process that requires production shutdowns and results in economic losses. Moreover, the inability to promptly identify valve leaks poses inherent safety risks. These operational challenges have created substantial demand for non-destructive testing (NDT) technologies for valve internal leakage detection. Among current solutions, acoustic emission testing has emerged as one of the most widely applied NDT methods for valve internal leakage monitoring [4,5,6].

Practical research has revealed that acoustic emission (AE) technology still faces multiple technical bottlenecks and challenges in valve internal leakage detection.

First, significant variations in the physical properties of oil and gas media lead to substantial differences in AE signals generated by different leaking media. This necessitates extensive experimental and sampling efforts. The propagation characteristics of such signals are influenced by multiple factors, including the physical state of the source, material type, environmental noise interference, and detection system response. Consequently, the acquired signals exhibit strong time-varying and non-stationary behaviors, requiring time–frequency joint analysis for dynamic feature extraction and identification.

Second, experiments demonstrate that under common leakage flow regimes, AE signals share certain similarities with environmental noise in both time and frequency domains. This often leads to false alarms in AE detection [7], making accurate separation between noise and target AE signals a persistent challenge.

Third, the diversity of valve types and specifications significantly increases the complexity of mathematical modeling. Different structural types of valves exhibit distinct internal flow path geometries and leakage pathways, resulting in markedly different AE signal characteristics. Even valves of the same type but different specifications produce significantly varied AE signals. This diversity demands specialized mathematical models that incorporate both valve structural parameters and medium characteristics, requiring substantial development effort.

Furthermore, experimental results show that when the leakage rate falls within lower ranges, the standard deviation of AE signals exhibits a significant positive correlation with the leakage rate. However, as the leakage rate increases, this correlation no longer holds. This nonlinear relationship makes it extremely difficult to establish a universally accurate descriptive model.

As illustrated in Figure 1, the realization of acoustic emission detection for valve internal leakage is confronted with four formidable technical challenges. Firstly, the signals manifest pronounced complexity when exposed to heterogeneous acoustic sources, varying media compositions, and diverse operational conditions, which significantly complicates signal feature extraction and analysis. Secondly, the detection process is inevitably plagued by various noise interferences, including ambient environmental noise and system-induced artifacts, thereby degrading the signal-to-noise ratio and posing obstacles to accurate signal interpretation. Thirdly, constructing an effective detection model presents inherent difficulties due to the non-stationary and nonlinear characteristics of acoustic emission signals, demanding sophisticated algorithms and techniques to capture the underlying patterns. Lastly, the relationship between the leakage rate and the signals is highly nonlinear, rendering traditional linear modeling approaches inadequate and necessitating the adoption of advanced nonlinear regression or machine learning methods for precise quantification and prediction.

In light of the aforementioned challenges and difficulties, the critical issues of how to process acoustic emission (AE) signals and select appropriate characteristic parameters and signal processing algorithms have become urgent problems requiring resolution. This paper systematically summarizes commonly used AE signal processing algorithms developed in recent years.

## 2. Overview of Acoustic Emission Detection Technology

Acoustic emission (AE) refers to the physical phenomenon where transient stress waves are generated by the rapid release of elastic energy when a material undergoes deformation or external forces. Each AE signal emitted from the source contains critical information about internal structural changes, defects, or state variations within the material.

The fundamental principle of AE testing technology involves using sensitive instruments to capture and process these AE signals. The process of acoustic emission detection is shown in Figure 2.

When there is a tiny defect inside the valve causing leakage, a pressure drop will occur near the area where the medium flows out due to the throttling effect. The medium will be ejected at high speed from the internal leakage position, causing disturbances in the air flow field near the internal leakage position and triggering intense turbulence. At the same time, shock waves may also occur [8]. The aerodynamic noise generated by the significant pressure changes in the flow field environment at the internal leakage position is transmitted to the medium outlet pipe wall in the form of mechanical waves along with the instantaneous elastic waves released by the tiny defect structure of the valve body under the action of stress, causing the leakage flow passage of the valve body to vibrate. In fact, the continuous radiation noise of intense turbulent pulsation and the sudden high-frequency cavitation pulses actually belong to the flow-induced noise phenomena in the field of pressure acoustics.

The leakage principle of the ball valve is shown in Figure 3. The left figure represents the normal working state of the ball valve, while the right figure shows the leakage state of the ball valve. The black arrows indicate the flow path of the fluid, and the positions pointed by the red arrows are the sealing areas of the ball valve. In the left figure, the fluid is confined within a certain range, while in the right figure, the fluid overflows from the sealing area.

Therefore, the following is a brief introduction to the theoretical research on flow-induced noise. In 1952, Lighthill innovatively proposed the Lighthill equation as follows. Its acoustic boundary conditions are shown in Figure 4 [10]:(1)$$\frac{\partial^{2}\rho^{\prime }}{\partial t^{2}}-c_{0}^{2}\frac{\partial^{2}\rho }{\partial x_{i}\partial x_{j}}=\frac{\partial^{2}T_{ij}}{\partial x_{i}\partial x_{j}}$$

In the equation, $\rho^{\prime }$ represents the density perturbation caused by sound pressure, defined as $\rho^{\prime }=\rho -\rho_{0}$, $c_{0}$ is the speed of sound, and $T_{ij}$ is the Lighthill stress tensor, defined as: $T_{ij}=\rho u_{i}u_{j}+\left(p^{\prime }-c_{0}^{2}\rho^{\prime }\right)\delta_{ij}-\tau_{ij}$. This model treats fluid motion noise as a quadrupole sound source (derived from the second-order derivative of the turbulent stress tensor $T_{ij}$), laying the theoretical foundation for aeroacoustics, but it is only applicable to free turbulent flow problems. In response to the impact of solid boundaries, Curle extended this equation in 1955. It incorporates the influence of solid boundaries, enabling the equation to more accurately describe the situation of acoustic emission generated by fluid flow in actual engineering, and providing a theoretical basis for analyzing acoustic emission in such scenarios [10]:(2)$$\rho^{\prime }\left(x,t\right)=\frac{1}{4\pi c_{0}^{2}}\frac{\partial^{2}}{\partial x_{i}\partial x_{j}}\int_{V}\frac{T_{ij}\left(y,t-\frac{\left|x-y\right|}{c_{0}}\right)}{\left|x-y\right|}d^{3}y+\frac{1}{4\pi c_{0}^{2}}\frac{\partial }{\partial x_{i}}\int_{S}\frac{p\left(y,t-\frac{\left|x-y\right|}{c_{0}}\right)n_{i}}{\left|x-y\right|}dS\left(y\right)$$

In the equation, p denotes the pressure acting on the solid boundary, $n_{i}$ represents the unit normal vector of the solid boundary surface, and S signifies the surface area of the solid boundary. V represents the fluid domain. The integration covers the entire fluid space, indicating the consideration of the contributions from all source points within the fluid. The extended formulation incorporates pressure fluctuations (dipole sources) on stationary boundaries, enabling the analysis of flow-induced noise in pipelines, casings, and other fixed-boundary systems. For the rigid body model, the type and area of its fluid noise are shown in Figure 5.

To further solve the problem of motion boundaries, in 1974, Goldstein, based on Green’s function method, unified the acoustic coupling mechanism between fluids and motion/complex boundaries, and eventually formed a generalized theoretical framework of flow-induced noise, providing a complete theoretical support for engineering noise prediction [13,14]. Therefore, ultrasonic sensors can be utilized to analyze the time–frequency signal characteristics and acoustic signal intensity of the signals generated on the outer wall of the downstream pipeline of the valve, thereby determining the internal leakage of the valve. Comparison of the Lighthill equation and the Curle extension as shown in the Table 1.

Building upon this theoretical foundation, researchers have further validated the feasibility of AE technology through experimental studies. As early as the 1950s, German scientist Kaiser made a seminal discovery while investigating the fatigue properties of metallic materials. He observed that when materials were subjected to loading, significant acoustic emission activity occurred as stresses reached or exceeded the historically experienced maximum stress level, whereas relatively minimal AE signals were detected below this threshold. This phenomenon, subsequently termed the Kaiser effect, established the theoretical basis for applying AE technology to valve internal leakage detection [15].

Figure 1 and Figure 6 present the schematic diagram of the Kaiser effect recognition method. In these diagrams, the vertical axis denotes the cumulative number of acoustic emission events, while the horizontal axis represents the stress exerted on the material, which is used to indicate the maximum stress value that the material has experienced. The figures illustrate that when the material is subjected to a stress level surpassing its historical maximum stress, a substantial number of acoustic emission events will take place. Consequently, this method is applicable for the non-destructive testing of internal leakage in valves. Numerous researchers have continuously advanced this technology, progressively improving its precision in non-destructive testing applications. The AE technology provides an effective means for quantitative leakage rate detection by capturing quadrupole and dipole acoustic source signals generated during valve leakage processes. Figure 7 shows the schematic diagram of the AE detection principle.

However, the acoustic emission signals obtained in actual detection are often accompanied by noise interference and nonlinear characteristics. Therefore, how to effectively extract leakage characteristics through signal processing methods has become a key issue. The following text will focus on discussing the research progress of processing methods for acoustic emission signals.

## 3. The Processing Method of Acoustic Emission Signals

### 3.1. Parameter Analysis Method

The parameter analysis method stands as one of the most classical and efficient approaches in acoustic emission (AE) signal processing. Its core principle involves extracting key time-domain characteristic parameters from signals—such as ring-down count, energy, and amplitude—to transform complex waveform information into quantifiable indicators [17], thereby enabling rapid assessment of material or structural damage states [18].

Prior to the 1970s, AE signal analysis primarily relied on time-domain parameters (e.g., ring-down count, amplitude) and analog filtering techniques. Even in modern research, scholars continue to employ parameter analysis methods for AE signal processing, demonstrating its enduring relevance in both fundamental studies and industrial applications.

In 2016, Fabian Lissek et al. [19] utilized acoustic emission (AE) technology for in-process monitoring during the abrasive waterjet cutting of carbon fiber-reinforced polymers (CFRP), employing parametric analysis of AE signals with a focus on the relationship between burst count, burst energy, and machining quality. Experiments involving linear cuts were conducted by varying water pressure, abrasive mass flow rate, and feed rate, with AE signals captured using an sensor. The results revealed significant correlations: increasing feed rate decreased burst count and increased energy per unit cut volume, consequently reducing cut quality, while increasing abrasive mass flow rate increased burst count and decreased energy per unit cut volume, thereby improving cut quality. High-quality cuts were characterized by high burst counts and low energy per unit cut volume; conversely, low-quality cuts exhibited low burst counts, high energy per unit cut volume, and were often accompanied by damage such as delamination or spalling. The characteristic parameters of acoustic emission are shown in Figure 8.

In 2024, Jing Xie et al. [20] used parametric analysis to process acoustic emission signals to detect valve leakage. An innovative two-step extraction method (MEL-GAN) was proposed, which extracts features in two steps. In the first step, the amplitude of the original acoustic emission signal is normalized, frames are divided, windows are added, and a discrete Fourier transform is performed. Then, Mel frequency conversion and filtering are carried out to obtain the Mel spectrum, and convolution is used to generate the two-dimensional feature map. The second step is to further extract the features of the first step using the GANomaly model, train the model with leak-free state data, and take the two differences in the model as new two-dimensional feature parameters. Through experiments, by using the K-means clustering algorithm to analyze this feature, it can effectively distinguish between the non-leakage and leakage states of the valve. Moreover, in the leakage state, the damaged leakage of the gasket can be identified with high precision, and the misjudgment rate of micro-opening leakage is 7.18%. This provides a new approach for valve leakage detection.

In 2025, Ahmad Braydi et al. [21] presented an innovative prognostic methodology for pipe defect detection in their paper “Innovative prognostic methodology for pipe defect detection leveraging acoustic emissions analysis and computational modeling integration.” This approach integrates acoustic emission (AE) analysis of bubble detachment with computational modeling. Parametric analysis of the acoustic signatures generated during bubble detachment from pipes revealed that their frequency and amplitude are highly sensitive to blockage levels and are influenced by parameters such as pipe inner diameter and gas flow rate. Leveraging Abaqus finite element simulations, the study systematically investigated the impact of various blockage characteristics—including thickness, length, and position—on acoustic wave propagation, extracting 16 key acoustic parameters encompassing both time-domain and frequency-domain features. These parameters were optimized to a 15-dimensional feature set using recursive feature elimination (RFE) for subsequent machine learning input. Experimental validation demonstrated that the method, utilizing classifiers such as Decision Trees and Random Forests, achieved 100% accuracy in detecting the existence, shape (configuration), and severity of blockages; furthermore, the model exhibited superior generalization capability for blockage thickness compared to shape.

When Yin et al. [22] were studying the failure characteristics of coal and rock under uniaxial loading conditions, they also adopted parametric analysis methods to process acoustic emission signals. After processing the coal samples retrieved from the site into homogeneous standard cylindrical specimens, uniaxial compression tests were conducted. Firstly, a preload of 2 kN was applied in a force value control mode of 0.5 kN/s, and then the loading was controlled by axial constant displacement at a rate of 0.1mm/min. During the test, sensors were used to synchronously collect characteristic parameters such as the amplitude of acoustic emission, ringing count, and event number, as well as stress–strain data. After collection, these acoustic emission parameters were combined with the stress-strain curve to analyze the parameter changes at different loading stages, thereby successfully revealing the failure evolution mechanism of coal and rock [23].

Li et al. [24] conducted feature extraction of the signal from both the time domain and the frequency domain dimensions, respectively. In the time domain, characteristic parameters such as standard deviation, maximum amplitude, crest factor, absolute mean value, root mean square, and energy were selected; In terms of the frequency domain, characteristics such as peak frequency, frequency centroid of the frequency band, skewness, and kurtosis were selected. Subsequently, the cross-entropy sorting method is used to systematically compare the extracted time-domain and frequency-domain features. From this, it is identified that the energy and frequency centroids are not suitable for pattern recognition. Based on this, it is determined that the four characteristics, maximum amplitude, absolute value mean, kurtosis, and peak frequency, will be used for the subsequent tests. The research team trained the artificial neural network model using a combined feature set containing the above four features, and the model demonstrated an outstanding estimation accuracy rate of 97.2%. When training is conducted using only the absolute mean value and peak frequency, the accuracy rate drops to 96.9%. However, if training is carried out only based on a single feature, the accuracy rate drops significantly to below 90%. His research results fully demonstrate that the comprehensive application of time-domain and frequency-domain characteristics can significantly reduce the occurrence of false alarms.

In valve internal leakage detection, the parameter analysis method has become a widely adopted tool for industrial field monitoring due to its real-time capability and computational efficiency. However, its inability to capture frequency-domain information limits diagnostic accuracy for complex faults, particularly in distinguishing between different leakage types. Consequently, it is often combined with time–frequency analysis methods as a front-end signal processing step to enhance detection reliability.

The industry urgently requires new processing techniques capable of rapidly identifying AE signal frequency characteristics to improve fault-type differentiation.

### 3.2. Time–Frequency Analysis

The time–frequency analysis method, combining time-domain and frequency-domain analysis, overcomes the deficiency of traditional parametric analysis methods in obtaining frequency-domain information, providing a new technical approach for acoustic emission signal processing.

#### 3.2.1. Fourier Transform

Fourier transform, as a classic frequency-domain analysis method, holds an important position in AE signal processing. Fleischmann et al. [25] first conducted spectral analysis of acoustic emission signals using Fourier transform in 1975 and revealed the high-frequency characteristics of material plastic deformation through transfer function correction, which pioneered the frequency-domain analysis. When Tang Boqiang studied the leakage acoustic emission signal in the gas valve, he used the fast Fourier transform (FFT) to convert the time domain acoustic emission signal into power spectral density (PSD), and found the significant peaks related to the leakage rate in the 40 kHz and 150 kHz frequency bands, which revealed the frequency domain energy distribution law of the leakage noise. Combined with Parseval’s theorem, the equivalence of the time domain signal energy and the frequency domain integral energy was verified [26]. His application of FFT not only supplements the deficiencies of the traditional parameter analysis method but also provides a certain quantitative basis for the frequency-domain diagnosis of valve internal leakage.

However, FFT is only applicable to stationary signals and cannot handle non-stationary transient AE signals such as turbulent noise. Its limitations have also given rise to more advanced time–frequency analysis techniques and promoted the development of subsequent time–frequency analysis techniques.

#### 3.2.2. Wavelet Transforms

It can be known from the above text that the Fourier transform analysis cannot solve the contradiction between time and frequency resolution. As a new time–frequency analysis method, wavelet analysis, with its multi-resolution feature, enables it to more effectively achieve the time-domain and frequency-domain analysis of signals. Wavelet has a wide range of applications in signal analysis and can be used for time–frequency analysis, separation of signal noise, identification and discrimination of signals, and calculation of fractal index, etc. The basic idea of wavelet transform is to analyze the signal by translating and scaling a set of finite energy mother wavelets $\left(\varphi \left(t\right)\right)$ to achieve adaptive time–frequency analysis [23].

The definition of wavelet packets: Let $\varphi \left(x\right)$ and $\psi_{0}$ be the scaling function and the wavelet function:(3)$$\psi_{0}\left(x\right)=\varphi \left(x\right),\psi_{l}\left(x\right)=\psi \left(x\right),$$(4)$$\psi_{2l}\left(x\right)=\sum_{k=-\infty }^{+\infty }h_{k}\psi_{l}\left(2x-k\right),\psi_{2l+1}\left(x\right)=\sum_{k=-\infty }^{+\infty }g_{k}\psi_{l}\left(2x-k\right)$$

The defined functions {ψn} are referred to as the wavelet packets associated with the scaling function ψ(x). The mathematical representation of the wavelet transform involves taking the inner product of the target signal x(*t*) with scaled and shifted versions of a base wavelet function ψ(*t*):(5)$$WT_{x}\left(a,\tau \right)=\frac{1}{\sqrt{a}}\int_{-\infty }^{+\infty }x\left(t\right)\psi^{*}\left(\frac{t-\tau }{a}\right)dt,a>0$$(6)$$WT_{x}\left(a,\tau \right)=\frac{\sqrt{a}}{2\pi }\int_{-\infty }^{+\infty }X\left(\omega \right)\psi \left(a\omega \right)e^{j\omega \tau }d\omega$$

Here, $X\left(\omega \right)$ and $\psi \left(\omega \right)$ represent the Fourier transforms of x(*t*) and ψ(*t*), respectively. The wavelet ψ(*t*) generates a set of basic functions, ψ, a, and τ(*t*), through scaling and translation operations [27].

The multi-resolution analysis of wavelet transform has excellent localization properties in both the time domain and the frequency domain, and it is one of the most widely used methods for front-end processing of acoustic emission signals at present. Wavelet packet is an extension of wavelet analysis, which can perform multi-level division of frequency bands and obtain combinations of arbitrary sub-band bandwidths. The high-frequency signals not analyzed by wavelet analysis can be deeply decomposed [28] to improve the time–frequency resolution [27,29,30].

The application research of wavelet analysis in the detection of internal leakage in valves shows that this method can effectively extract the high-frequency characteristics of the leakage signal and significantly improve the detection accuracy. However, the selection of wavelet basis functions has an important influence on the analysis results. The team of Juhani J from Malaysia compared and analyzed the processing effects of four wavelet basis functions, namely, db, sym, bior, and coif, on the AE signal of valve internal leakage. Based on the evaluation of the ratio of energy to information entropy, the research shows that the db wavelet is the optimal choice [31].

Sim et al. [32] aimed at the valve leakage problem of reciprocating compressors. They used the Daubechies 9(db9) wavelet basis to perform three-layer discrete wavelet decomposition on the AE signal and extracted 112.5–225 kHz as the characteristic frequency band. By calculating the RMS value and entropy value of this frequency band, the background noise interference was significantly reduced, and it was found that the entropy value was more sensitive to the intermittent impact of viscous valves. The experimental results are shown in Figure 9.

This result is consistent with the conclusion of db wavelet optimization proposed by Juhani’s team, further verifying the importance of wavelet basis selection for industrial signal processing.

Arnab Gupta et al. [33] proposed using wavelet decomposition of ultrasonic signals to identify the arrival times of longitudinal and bending waves. After collecting ultrasonic signals, a continuous wavelet transform was performed using real-valued Morlet wavelets to decompose the signals into different frequency components. High-frequency components (e.g., 310 kHz) were used to identify the arrival time of longitudinal waves, while the element-wise product of low-frequency components (70–270 kHz) was employed to determine the arrival time of bending waves. The results show that this algorithm is effective for signals with different noise levels and source distances.

From the research mentioned above, it can be seen that by using the time–frequency analysis method with wavelet transform as the core to process the acoustic emission signal through the combination of time and frequency resolution, the limitations of traditional FFT in dealing with non-stationary acoustic emission signals are effectively solved, providing a high-precision feature extraction method for scenarios such as valve internal leakage and material damage.

#### 3.2.3. Other Adaptive Signal Decomposition Methods

Empirical Mode Decomposition (EMD), Local Mean Decomposition (LMD), and Variational Mode Decomposition (VMD) represent three adaptive signal processing techniques widely used for acoustic emission (AE) signal analysis. EMD decomposes signals into intrinsic mode functions (IMFs) through an iterative sifting process of extreme-point envelopes, though it suffers from mode-mixing issues. LMD improves upon EMD by calculating local means and envelopes to produce product functions (PFs), achieving better component separation. The most advanced among them, VMD, employs a variational optimization framework to decompose signals into a preset number of band-limited IMFs in the frequency domain. By minimizing each mode’s bandwidth, VMD effectively avoids mode mixing while demonstrating superior noise robustness. Although all three methods are data-driven and adaptive, VMD’s rigorous mathematical foundation and frequency-domain processing capability make it particularly suitable for analyzing complex AE signals encountered in valve leakage detection.

In their research, Gao et al. [34] systematically compared the performance differences of three signal decomposition methods, namely, EMD, LMD, and VMD, and experimentally verified the excellent performance of VMD. Experimental results demonstrate distinct performance differences among these decomposition methods when processing AE signals from nuclear power plant gate valves. Traditional EMD exhibits significant mode mixing during decomposition, while LMD’s improved local mean processing only partially mitigates this issue and remains susceptible to endpoint effects. In contrast, VMD’s variational optimization framework and frequency-adaptive segmentation capability not only effectively suppress mode mixing but also yield IMF components with superior time–frequency stability. Quantitative analysis reveals that the primary IMF component selected through a K–L divergence measurement shows significantly higher correlation with the original signal in VMD processing compared to other methods. These advantages collectively enable the VMD-based fault diagnosis model to achieve an average accuracy of 96.375%, conclusively validating VMD’s superiority for complex industrial signal processing applications.

Li et al. [35] also developed an adaptive noise reduction method based on VMD, which automatically screens the leakage-related modal components by defining the “signal clarity index”, which effectively solves the interference problem of mixed noise that is difficult to handle by traditional methods, and the signal-to-noise ratio is improved by up to 9 dB [35]. Actual tests show that this method controls the pipeline leakage location error within the range of 0.17–2.18%, which is significantly superior to the wavelet and EMD methods, providing a reliable solution for pipeline leakage detection in complex environments.

In 2022, Li Zhixing et al. [36] proposed an improved VMD method for pipeline leakage detection. The optimal number of decomposition layers was determined by the correlation coefficient method, and the best penalty factor was selected based on the kurtosis value. This method effectively separates the modal components with the highest correlation to the original signal. Combined with Hilbert envelope spectrum analysis, the leakage characteristic frequencies are successfully extracted, significantly improving the signal-to-noise ratio. Studies show that the optimized VMD has excellent feature separation ability in complex industrial signal processing.

In 2020, Igor Rastegaev et al. [37] proposed an acoustic emission (AE) evaluation method based on time–frequency analysis for the in-process monitoring and mechanism identification of sliding wear processes. Addressing the challenge of extracting useful information amidst multi-source noise and spurious AE signals within friction systems, the study employed a threshold-free continuous AE acquisition technique combined with two types of broadband filtering algorithms and three clustering algorithms. Validated through standardized friction tests such as four-ball and pin-on-disc configurations, the research discovered that AE signals generated by different wear mechanisms exhibit distinct power spectral density characteristics. The combination of filtering algorithm F1.1 and clustering algorithm K1 proved most effective, successfully differentiating three typical AE signal clusters: circle type (indicating abrasive-dominated wear), upward-pointing triangle type (indicating local adhesion and early-stage scoring), and inverted triangle type (indicating severe scoring/welding). The time–frequency analysis successfully correlated these AE signal clusters with the evolution of wear morphology, enabling highly sensitive detection of failure-critical transition points [37].

However, as industrial inspection scenarios grow increasingly complex, reliance solely on time–frequency analysis still faces challenges such as experience-dependent feature selection and limited generalization capability. To address these limitations, modern intelligent detection methods—including deep learning and optimization algorithms—are emerging as pivotal solutions to overcome traditional technical bottlenecks. These approaches leverage adaptive feature learning and nonlinear modeling to automatically extract hidden patterns from acoustic emission signals. More significantly, through end-to-end training, they enable intelligent mapping from raw data to fault diagnosis, thereby substantially enhancing the engineering applicability of acoustic emission technology.

### 3.3. Nonlinear Dynamic Analysis Method

While time–frequency analysis methods can effectively characterize the time-varying features of acoustic emission (AE) signals, they still exhibit certain limitations when processing non-linear and non-stationary signals. To address these challenges, researchers have turned to nonlinear dynamics approaches, which extract complex dynamic characteristics embedded within the signals, offering novel solutions for AE analysis. The following two representative cases demonstrate innovative breakthroughs achieved by this methodology in different application scenarios.

The Zhao team proposed two innovative improvements in response to the limitations of the traditional multi-scale entropy (MSE) in acoustic emission signal processing. The first one is the local maximum multi-scale entropy (LMMSE), which constructs envelope features by extracting the maximum value of signal segments, effectively avoiding information overlapping caused by mean operations. The second is the extended multi-scale entropy (EMSE), which adopts negative scale factors and anti-aliasing filtering techniques to achieve nonlinear expansion of the signal frequency band. Experiments show that the improved method increases the feature recognition accuracy by more than 30% (up to 92%) in bearing fault diagnosis and has stronger noise robustness [38]. These methods provide new ideas for extracting non-stationary features of transient impulse acoustic emission signals and are particularly suitable for early fault diagnosis under small sample conditions.

Meanwhile, Agletdinov et al. [39] utilized recursive quantization analysis methods, through phase space reconstruction and Shannon entropy calculation, to establish a quantitative correlation between acoustic emission characteristics and metal dislocation dynamics, achieving high-precision prediction of the critical point of material plastic instability (with an error of <10%). These achievements not only expand the application dimensions of acoustic emission technology but also provide a new analytical paradigm for the condition monitoring of complex systems.

### 3.4. Traditional Machine Learning

With the escalating complexity of industrial systems, conventional acoustic emission (AE) signal processing methods have revealed growing limitations. In recent years, intelligent detection approaches—through the integration of machine learning, deep learning, and optimization algorithms—have endowed AE technology with robust adaptive feature extraction and high-dimensional pattern recognition capabilities, significantly enhancing fault diagnosis accuracy and generalizability.

These data-driven intelligent methods eliminate reliance on manual feature engineering by directly learning deep representations from raw signals. For instance:(1)Convolutional Neural Networks (CNNs) autonomously extract discriminative time–frequency features;(2)Long Short-Term Memory (LSTM) networks effectively model complex temporal dependencies between signals and damage states.

Furthermore, the incorporation of transfer learning and data augmentation techniques has substantially improved system robustness in industrial environments, successfully addressing critical challenges such as small-sample learning and noise interference. The following sections will systematically illustrate, through representative case studies, how these intelligent methods are facilitating a paradigm shift in AE detection—from “experience-dependent” to “autonomous decision-making” methodologies.

In 2025, Ayrat Zagretdinov et al. [40] proposed a pipeline leakage detection method based on the fractal characteristics of acoustic emission (AE) signals, utilizing the Hurst exponent (H-value) for leakage identification. The study employed R/S (Rescaled Range) analysis and Detrended Fluctuation Analysis (DFA), nonlinear dynamics techniques, to process pipeline AE signals, revealing that leak-free signals approximate deterministic processes, whereas leakage-induced turbulent fluctuations impart anti-persistent behavior to the signals; furthermore, variations in leak aperture exhibited minimal influence on the H-value. Computational Fluid Dynamics (CFD) simulations confirmed the scale-invariance of turbulent vortex frequencies across different leak apertures, explaining the stability of the H-value. Spectral analysis indicated that the dominant frequency band governing the fractal structure concentrated within 0–2 kHz, with acoustic waves in this range propagating up to 600 m, making it suitable for long-distance monitoring. Field validation on pipelines with diameters of 50 mm and 100 mm demonstrated the method’s reliability: the H-value during leakage fell significantly below the leak-free confidence interval (α = 0.05). Compared to traditional spectral methods, this nonlinear analysis circumvents the requirement for signal stationarity, offering a novel approach for leakage monitoring in complex pipeline networks [40].

#### 3.4.1. The Application of Support Vector Machine

Support Vector Machine (SVM) has become one of the preferred algorithms for acoustic emission signal classification due to its excellent performance in the case of small samples. In terms of algorithm optimization, Gou Yunfeng from Chongqing University proposed an SVM model optimized based on the Sparrow Search Algorithm (SSA). By comparing multiple kernel functions, it was found that the SVM with the RBF kernel function had the best classification effect. Further, the SVM parameters of the RBF kernel function were optimized using the SSA algorithm, and the SSA-SVM model was constructed with the classification accuracy rate of SVM as the fitness function. The classification accuracy rate of this model reached 98.148%, which was superior to the SVM model optimized by the grid search method and could effectively identify the internal leakage sound signal of the valve [41].

In 2025, Gong Jiale et al. [42] further improved the SSA-SVM model in the research on the diagnosis of internal leakage in valves. First, construct an SVM model with valve sound and temperature data as input and leakage degree as output, and improve the multi-classification ability by using the one-to-many method and radial basis function. Then, SSA is used to simulate the behavior of sparrows to optimize the hyperparameters of SVM. It classifies sparrows into discoverers, adders, and thieves, and balances global and local searches through different position update strategies. During the optimization, the parameters c and gamma of the SSA-SVM model are dynamically adjusted. Extensive search is conducted in the early stage of training, a better solution is found in the middle stage, and continuous exploration is carried out, and further optimization is performed in the later stage to prevent local optimum [43,44]. The comparison results show that the accuracy rate of the SSA-SVM model is 99%, the precision rate is 98%, and the recall rate is 98.5%. Although the diagnosis time increases by 30 s, in scenarios with high accuracy requirements, it can more effectively identify valve leakage, reduce false alarms and missed alarms, and improve the reliability and accuracy of diagnosis.

In the fault diagnosis of reciprocating compressor valves, Sim et al. [32] extracted the time–frequency characteristics of acoustic emission signals through DWT and classified the valve states in combination with SVM, which were divided into four states: normal, viscous, single leakage, and double leakage. By adopting the RBF kernel and the “one-to-one” multi-classification strategy, the overall accuracy rate of the model exceeds 90%, with the highest reaching 98.8%. It is found that it is significantly superior to the KNN algorithm, especially performing better when distinguishing between viscous and leakage states. Then, Zhang et al. [7] combined SVM with KNN to construct a typical multi-stage diagnostic framework: Firstly, DWT was used to extract the time–frequency characteristics of the acoustic emission signal, and SVM was used as the core classifier to achieve the accurate identification of the valve status, with an accuracy rate of 98.4%. When working in coordination with KNN and regression models, fault classification and leakage quantification can be accomplished simultaneously. Compared with the traditional single SVM model, this strategy of feature extraction and multi-algorithm fusion significantly improves the robustness and engineering applicability of the system, providing a new idea for the intelligent diagnosis of industrial equipment. This achievement also confirms the characteristic that SVM can better balance the model complexity and classification performance when working in collaboration with other algorithms.

In terms of feature engineering, Mysorewala et al. [45] increased the detection accuracy of SVM from 51% to 88% through multi-sensor data fusion. Liu et al. [46] achieved a binary classification accuracy rate of 99.3%. Yang Liu achieved a recognition rate of 98% by combining EMD–ApEn–PCA feature engineering. These studies collectively demonstrate that Support Vector Machines (SVM), through kernel function techniques and feature optimization, can effectively handle the nonlinear characteristics of pipeline leakage acoustic emission (AE) signals. The method consistently maintains stable accuracy rates above 90% across diverse leakage scenarios, establishing itself as a reliable technical solution for industrial detection applications.

#### 3.4.2. Optimization and Extension of Traditional Machine Learning Methods

In the research on the integration of signal processing and machine learning, different teams have proposed innovative solutions for various industrial inspection requirements. Xie et al. [47] extracted the characteristics of acoustic emission signals through wavelet packet decomposition and achieved multi-sensor information fusion by combining the SVM classifier and D–S evidence theory, increasing the accuracy of pipeline leakage detection to 99.67%. In a parallel development, Li-Ping Liang’s research team proposed an innovative approach for valve internal leakage detection by combining wavelet scattering transform (WST) with ensemble learning. Their methodology first decomposed acoustic emission signals using WST to generate stable time–frequency representations, from which the first three orders of scattering coefficients were extracted as primary features. The Relief-F algorithm was then employed to select the most discriminative features while eliminating redundancy. For classification, an AdaBoost.M1 ensemble model was constructed using CART decision trees as base classifiers, which iteratively adjusted sample weights to prevent overfitting—a critical advantage for small-sample industrial applications. This integrated approach demonstrated remarkable performance in distinguishing between normal, incipient, and severe leakage states, achieving 15–20% higher accuracy compared to conventional single-classifier models while maintaining robust performance across varying operating conditions. The success of this framework lies in its synergistic combination of WST’s signal representation stability, Relief-F’s feature selection capability, and AdaBoost’s adaptive boosting mechanism, offering a reliable solution for valve condition monitoring under data-limited scenarios commonly encountered in field applications [48].

Ma et al. [49] proposed an acoustic emission (AE) signal processing method based on an improved sparrow search algorithm (ISSA)-optimized random forest (RF) model. The ISSA was improved through opposition-based learning and random walk strategies to enhance global search capability and avoid local optima issues inherent in traditional SSA. Six key features strongly correlated with leakage flow were selected from time–frequency domain characteristics using Pearson correlation coefficients, forming the ISSA-RF hybrid model. Experimental results demonstrate that the method achieved a prediction coefficient of determination (R^2^) of 0.8940 on 300 multi-condition datasets, with a mean absolute error (MAE) of 8.22 L/min, significantly outperforming conventional SVR and unoptimized RF models. This provides a high-precision, non-intrusive solution for industrial pipeline leakage monitoring.

Then, neural network technology was gradually applied to the back-end optimization of acoustic emission signals. BP neural network was a typical representative in the early stage. In the aspect of pipeline leakage detection, the research of Jialin Cui and Jinbo Du [50] measured the wave velocity through lead breaking experiments and comprehensively used CEEMDAN mode decomposition and HB weighted cross-correlation algorithm for preliminary positioning. It innovatively introduced the BP neural network for error correction, significantly improving the positioning accuracy. The average error was reduced from 6.09% to 3.07%, achieving efficient and precise positioning of pipeline leakage sources in a complex noise environment.

Shi Mingliang’s team proposed a hybrid model for detecting internal leakage in ball valves, which combines wavelet packet denoising with the Sparrow Search Algorithm (SSA) to optimize the BP neural network. After preprocessing the acoustic emission signal with wavelet packet denoising in the research, the key characteristic parameters were extracted, and the Sparrow Search algorithm (SSA) was innovatively used to optimize the initial weights and thresholds of the BP neural network.

The experimental results show that the prediction error of this SSA–BP hybrid model is less than 6%, and its performance is significantly improved compared with the traditional BP neural network. It effectively solves the problem that the BP network is prone to falling into a local optimum [9], providing a more accurate intelligent diagnosis scheme for the leakage detection of industrial valves. The experimental process of the SSA-BP neural network is shown in Figure 10. 

Gao et al. [34] proposed a machine learning-based acoustic emission signal processing method for fault diagnosis of nuclear power plant motorized gate valves. As described in the previous section, after preprocessing the AE signals using VMD, they ultimately employed an improved Sparrow Search Algorithm (ISSA) to optimize the parameters of a random forest (RF) model, constructing a VMD–MDI–ISSA–RF fault diagnosis model. Experimental results demonstrate that this method achieves an average diagnostic accuracy of 96.375% for internal leakage and other faults, confirming its effectiveness and robustness in complex, noisy environments.

### 3.5. Deep Learning Method

Deep learning technology has shown significant advantages in the field of pipeline leakage detection, and its adaptive feature extraction ability has greatly improved the detection accuracy and robustness. According to the signal processing methods and the characteristics of the network architecture, the existing research can be classified into the following categories.

#### 3.5.1. Deep Learning Methods Based on Time–Frequency Feature Extraction

Zhou Wen’s team innovatively combined wavelet packet decomposition and deep belief network (DBN) in the detection of valve internal leakage [51]. By analyzing the energy distribution characteristics of the acoustic emission signal in each frequency band, it is found that the leaked signal has specific frequency-domain aggregation characteristics. The DBN model adopting the “pre-training-fine-tuning” strategy achieves the optimal evaluation effect when using time-domain characteristic parameter input, with the smallest mean square error.

In 2023, the team led by Zhang Zhiyuan and Xu Changhang from China University of Petroleum innovatively proposed an acoustic emission signal processing framework based on MFCC–LSTM for leak detection in gas–liquid two-phase flow pipelines [7].

The signal processing methodology employed in this system is delineated in Figure 11. Subsequent to preprocessing and time–frequency analysis, the signal proceeds to the MFCC-LSTM processing phase. In this research, the LSTM module is structured with an input layer, an LSTM layer, a fully connected layer, a SoftMax layer, and an output layer. Within the LSTM module, four distinct labels are utilized for discerning leakage sizes, each corresponding to varying widths of leakage apertures, along with two leakage state labels, specifically “normal” and “leak.” The MFCC input layer is configured with dimensions of 181 × 96, and the number of nodes in the fully connected layer is determined by the number of classification categories and the time steps of the input data.

This study introduces the Mel frequency cepstral coefficient (MFCC) in the field of speech recognition into acoustic emission signal processing for the first time. The flowchart of SSA-MFCC feature extraction is shown in Figure 12. By extracting 96-dimensional time–frequency features through a 24-layer Mel filter bank and conducting timing modeling in combination with the LSTM network, end-to-end intelligent diagnosis is achieved. The internal structure of the LSTM neuron is shown in Figure 13. The experimental results show that the leakage detection accuracy of this method under 1152 complex working conditions reaches 98.4%, which is more than 30% higher than the traditional methods. Among them, the recognition rate of crack leakage is as high as 100%. This research, through the integration of interdisciplinary technologies, provides an innovative solution with high precision and strong robustness for non-destructive testing of industrial pipelines.

#### 3.5.2. Convolutional Neural Network Method Based on Graphical Processing

Shukla and Piratla [52] constructed a CNN model based on the improved AlexNet architecture to process the scale graph of pipeline vibration acceleration signals, and the accuracy rate in PVC pipeline leakage detection exceeded 95%. In their latest study, Du’s team [53] employed bispectral analysis to transform signals into two-dimensional time–frequency images, combined with a convolutional block attention module (CBAM-CNN) to achieve end-to-end diagnosis. This method dynamically focuses on key features through channel-spatial dual attention mechanisms, achieving 98.87% classification accuracy for composite material damage, with recall rates reaching 100% for severe damage cases. This framework can directly process raw waveforms, significantly enhancing nonlinear feature extraction capabilities, and provides an effective solution for transient acoustic emission signal diagnosis. Song et al. [54] also innovatively designed a three-layer convolutional neural network structure based on large convolutional kernels to address the interference problem of fluid noise inside the pipeline on the leakage acoustic emission signal. This network effectively captures the long-term characteristics of the signal through a large receptive field, successfully achieving an efficient distinction between the leaked signal and the background noise. It can still maintain a recognition accuracy rate of more than 93% under the condition of strong noise interference. This research achievement further validates the significant advantages of deep learning in pipeline leakage detection: its adaptive feature extraction ability not only greatly improves the accuracy of signal analysis but also demonstrates outstanding leakage identification and classification performance under complex working conditions [55].

#### 3.5.3. Semi-Supervised and Unsupervised Learning Methods

Cody et al. [56] proposed a semi-supervised method combining CNN and variational autoencoder (VAE) and used the acoustic signal spectrograms collected by hydrophones for detection. When detecting a leakage of 0.25 L/s, the accuracy rate of this method reaches 97.2%. It performs well when dealing with a small amount of leakage data and effectively detects anomalies through reconstruction errors. Unsupervised learning has shown unique potential in acoustic emission signal processing. Mishra et al. [57] proposed a pipeline leakage detection method based on the Growing Neural Gas network (GNG), which overcomes the dependence of traditional supervised learning on label data through dynamic topological modeling. This method first extracts the time-domain (mean, kurtosis) and frequency-domain features (frequency center, RMS frequency) of the acoustic emission signal, then uses the GNG algorithm to construct the feature space graph of the health state and finally generates the pipeline health index (PHI) by calculating the Euclidean distance between the test sample and the nearest node. Experiments show that this index has a sensitivity of 96.5% for leaks as small as 0.3 mm under pressures of 2 bar and 7 bar, and the leakage size has a monotonically increasing relationship with the PHI value. Compared with GMM and SVDD, the dynamic node growth mechanism of GNG significantly improves the adaptability of the model to complex working conditions, providing a new idea for the problem of label scarcity in industrial scenarios.

#### 3.5.4. Cross-Domain Adaptation Method

The time–frequency convolutional neural network (TFCNN) model of Guo et al. [58] conducts time–frequency analysis on the leaked signal. Under different signal-to-noise ratio conditions, this model shows high detection accuracy and stability, with an average accuracy rate of 98%, and it can even reach 99% in practical applications. Furthermore, the TFCNN model based on transfer learning also shows good adaptability on different urban datasets.

Jia Hu et al. [59] conducted research on the leakage fault diagnosis of diesel engine exhaust valves. They collected acoustic emission signals at specific time intervals and input them into a one-dimensional convolutional neural network (1D_CNN) to extract discriminative features. They also constructed fault diagnosis models using three domain adaptive algorithms, namely, MDD, DANN, and DAN, for comparison.

1D_CNN classifies features after extracting them through operations such as convolution; MDD adds a classifier and utilizes the gradient inversion layer to reduce the domain distribution differences; DANN enables the feature extractor to learn domain-invariant features through adversarial training; DAN measures the inter-domain differences by means of the maximum mean difference of the kernel. The Figure 14. shows their grid structures:

The experimental results show that this method has a high accuracy rate in cross-working condition and cross-machine type fault diagnosis. Among them, the MDD algorithm has the best overall performance and effectively enhances the generalization ability of the fault diagnosis model. These research achievements demonstrate the outstanding performance of deep learning in the field of fault detection from different perspectives, providing important references and technical support for related engineering applications.

Sun Jiedi et al. [60] optimized the structure of the traditional dense connection network and proposed a lightweight neural network model based on improved dense blocks. While significantly reducing the number of parameters, it can still achieve a leakage aperture identification accuracy rate of 96.59%, providing a more efficient solution for industrial site deployment. Yao et al. [61] proposed an innovative method integrating energy mode clustering and feature reconstruction, 1D-CNN (FAE-1D-CNN), aiming at the problems of small samples, noise interference, and feature loss of the sound signal of natural gas pipeline leakage. The high and low energy mode features were separated through mean clustering. The correlation function is adopted respectively to enhance the weak features and the Butterworth filter to suppress the high-frequency noise. Subsequently, the local spatial features were reconstructed through the extended structure feature encoder (FAE), and, finally, a fault identification accuracy rate of 95.17% was achieved on the GPLA-12 dataset, providing an efficient solution for pipeline leakage detection in complex industrial environments.

### 3.6. Summary of Acoustic Emission Signal Processing Methods

Based on the above content regarding the processing methods of acoustic emission signals, several techniques are summarized and analyzed to identify their distinct characteristics, resulting in the following Table 2. 

## 4. Concluding Remarks

In summary, acoustic emission (AE) detection technology has made significant progress in the field of non-destructive testing for valve internal leakage, particularly through its deep integration with wavelet transform and intelligent algorithms, demonstrating strong potential for further development. The combination of AI-driven algorithms and AE-based detection has enabled automation and intelligent decision-making, spanning from data processing to fault diagnosis.

The application of various machine learning models, including Support Vector Machines (SVM), Neural Networks, Sparrow Search Algorithm (SSA), and Gaussian Process Regression (GPR), has not only improved the accuracy and adaptability of diagnostic models but also endowed them with self-learning and optimization capabilities. These models can automatically extract patterns from large datasets, accurately identify valve leakage states, and assess leakage severity, significantly reducing manual intervention and misjudgment risks while enhancing detection efficiency and reliability.

However, although many achievements have been made so far, there is still broad room for development in this field. On the one hand, it is necessary to further explore more efficient and intelligent algorithms or optimize and improve the existing algorithms to cope with more complex actual working conditions and diverse valve types and continuously improve the detection accuracy and speed. On the other hand, efforts should be made to enhance the integration of multiple technologies, such as combining the Internet of Things and big data, to achieve remote real-time monitoring, data sharing, and in-depth analysis of valve internal leakage, and to build a more complete intelligent monitoring system. At the same time, we should intensify research on practical application scenarios in different industries, promote the wide application of AE detection technology and intelligent algorithms in more fields, and provide a more powerful guarantee for the safe and stable operation of industrial production.

## Figures and Tables

**Figure 1 sensors-25-04487-f001:**
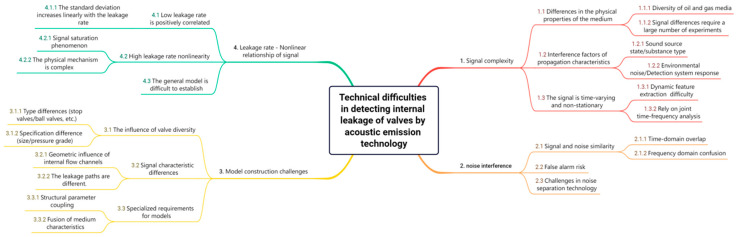
Technical challenges in detecting valve internal leakage using acoustic emission technology.

**Figure 2 sensors-25-04487-f002:**
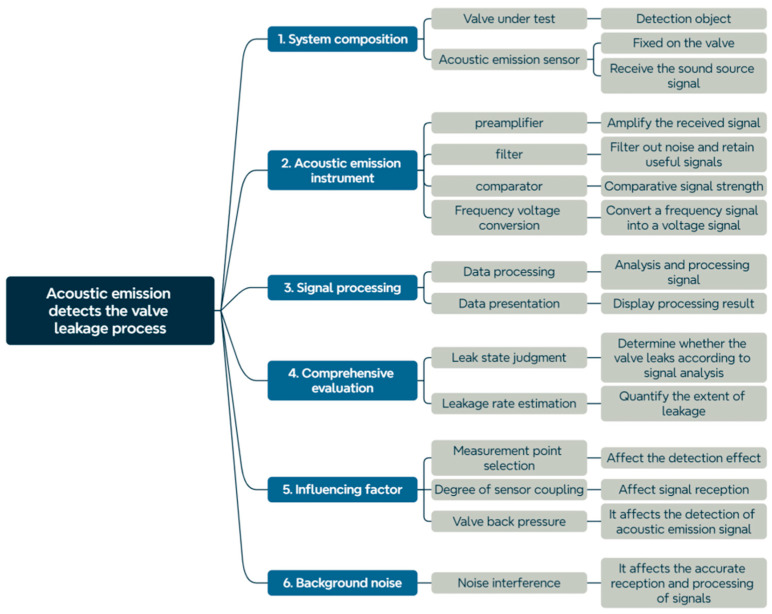
Process of valve internal leakage detection using acoustic emission testing technology.

**Figure 3 sensors-25-04487-f003:**
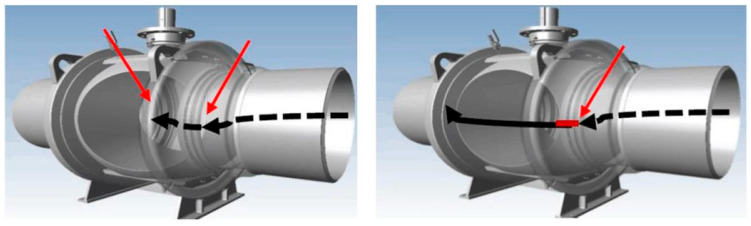
Schematic diagram of the ball valve leakage principle [9].

**Figure 4 sensors-25-04487-f004:**
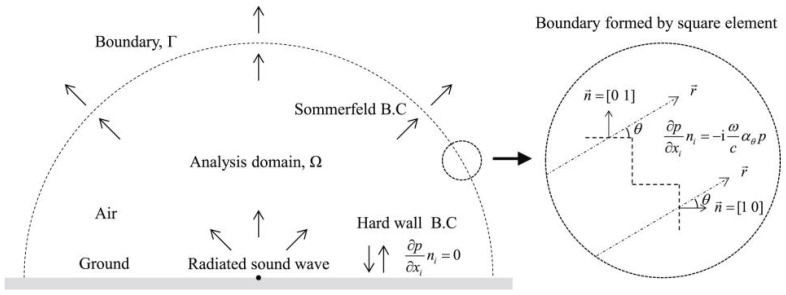
Acoustic boundary conditions of the semi-circular analysis domain based on the Lighthill formula [11].

**Figure 5 sensors-25-04487-f005:**
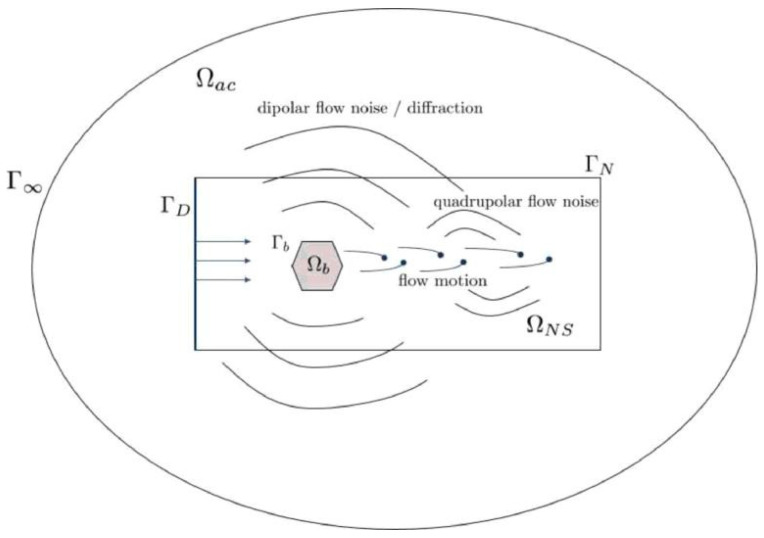
The dipole contribution of the rigid body *b* corresponds to the diffraction of the quadrupole flow noise generated at the tail passing through the rigid body [12].

**Figure 6 sensors-25-04487-f006:**
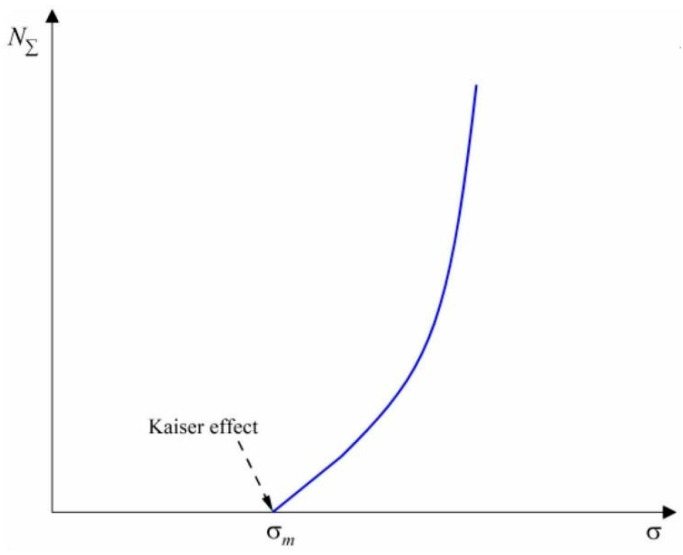
Kaiser Effect Identification Method: The activity of AE increases sharply [16].

**Figure 7 sensors-25-04487-f007:**
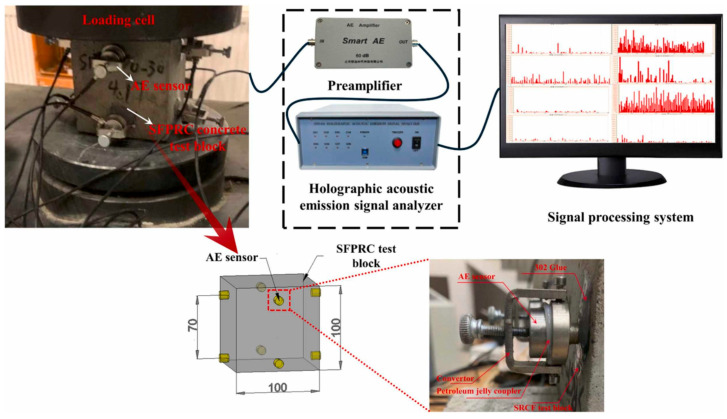
Schematic diagram of the AE detection system [17].

**Figure 8 sensors-25-04487-f008:**
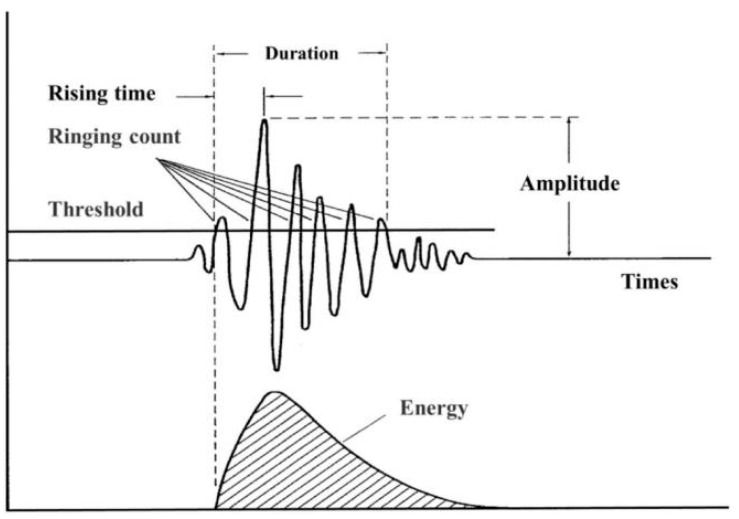
Characteristic parameters of AE [17].

**Figure 9 sensors-25-04487-f009:**
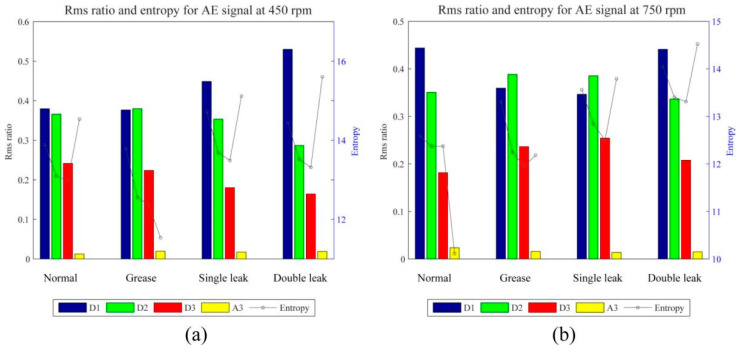
(**a**) Comparison of the root mean square ratio and entropy of each valve condition and decomposition level at 450 rpm and (**b**) 750 rpm [32].

**Figure 10 sensors-25-04487-f010:**
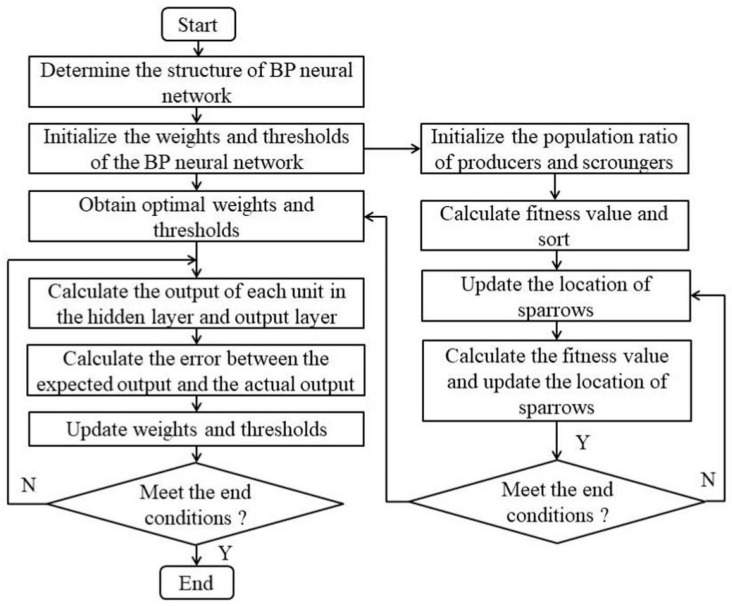
Flowchart of the SSA–BP neural network [9].

**Figure 11 sensors-25-04487-f011:**
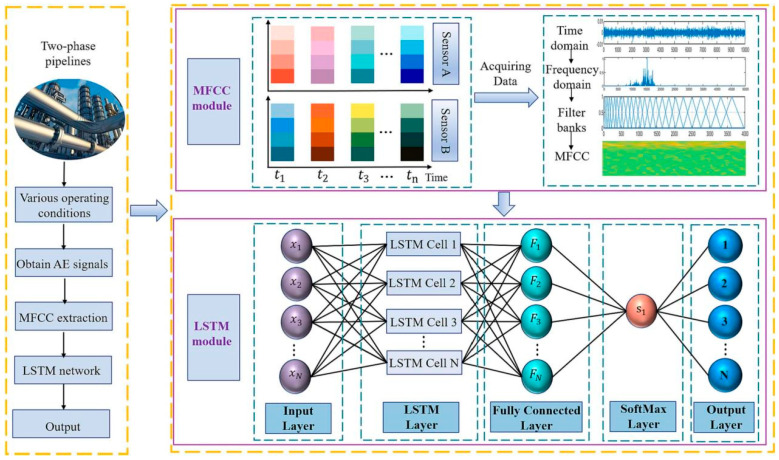
MFC–LSTM framework [7].

**Figure 12 sensors-25-04487-f012:**
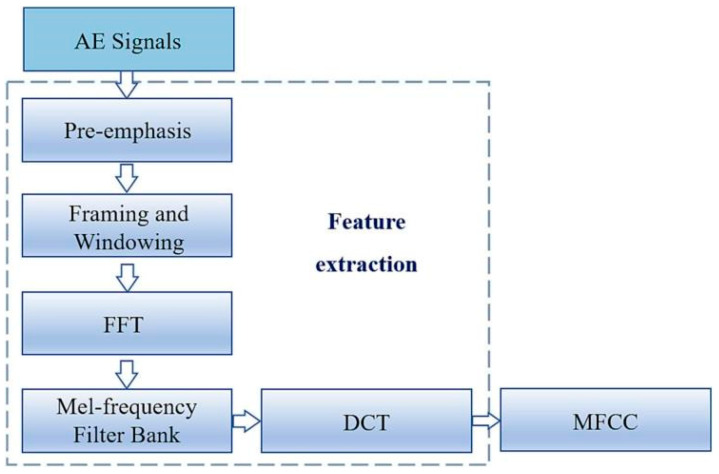
Flowchart of MFCC feature extraction [7].

**Figure 13 sensors-25-04487-f013:**
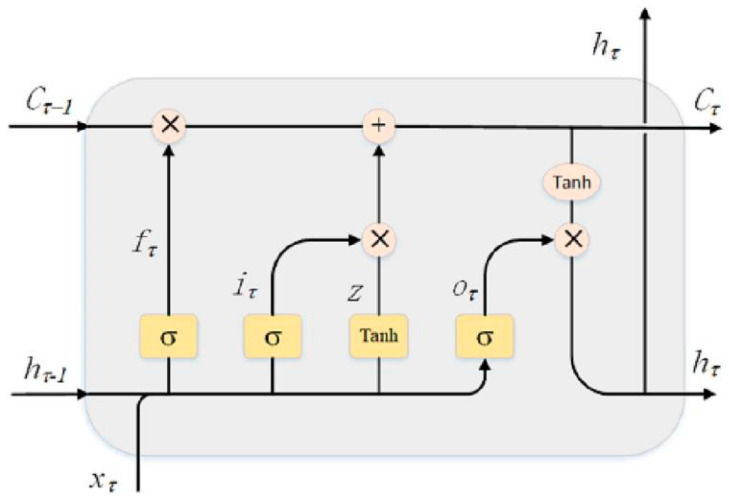
Internal structure of LSTM neurons [7].

**Figure 14 sensors-25-04487-f014:**
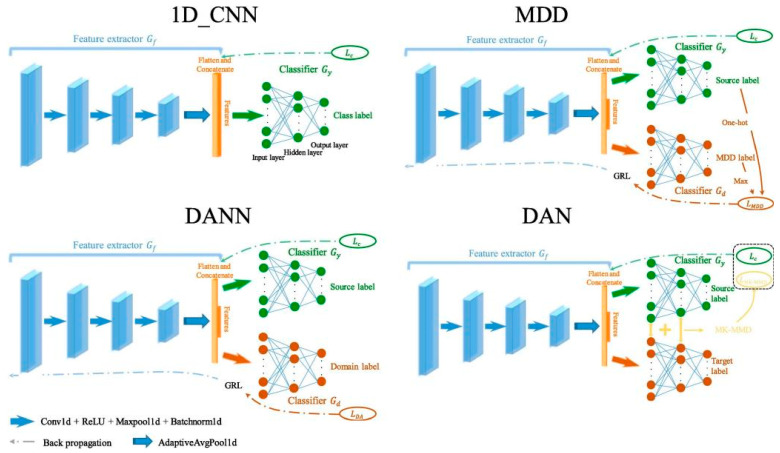
Network structures of D_CNN, DAN, DANN, and MDD [59].

**Table 1 sensors-25-04487-t001:** Comparison of the Lighthill equation (quadrupole sound source) and the Curle extension (dipole sound source).

Comparison Dimension	The Lighthill Equation (Quadrupole Sound Source)	Curle Extension (Dipole Sound Source)
Physical meaning	It is generated by the turbulent stress (velocity pulsation) within the fluid, with the sound source symmetrically distributed and no net force acting on it	It is generated by the pressure pulsation at the boundary between the fluid and the solid. The sound source is asymmetric, and there is a net force effect
Mathematical form	In the previous text, (1)	In the previous text, (2)
Sound source type	Quadrupole (two pairs of inverse force couples, similar to “dual speaker” radiation)	Dipole (a pair of reverse forces, similar to the vibration of a “single speaker”)
Acoustic radiation efficiency	low	high
Typical application scenarios	Free turbulent noise (valve jet noise)	Solid boundary noise (such as pipe wall vibration, valve internal leakage, and valve seat collision noise)
Schematic diagram description	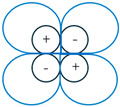	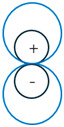

**Table 2 sensors-25-04487-t002:** General table of acoustic emission signal processing algorithms and optimization algorithms.

Technology Category	Processing Algorithm/Model	Optimization Algorithm Combination	Application Scenarios	Performance Index	Performance Index
Parameter analysis method	Time domain parameters (ringing count, energy)	No.	Rapid screening of valve status	Has high real-time performance but low accuracy	[17,20,22,24]
	Frequency domain parameters (peak frequency, centroid)	No.	Identification of leakage frequency bands	The energy distribution in the frequency domain is significant	[20,26]
	Mel–GAN two-step feature extraction	No.	Classification of Valve leakage	The misjudgment rate of micro-opening leakage was 7.18%	[20]
Time–frequency analysis	Short-time Fourier Transform (STFT	No.	Preliminary analysis of non-stationary signals	The time–frequency resolution is limited by the window length	[25]
	Continuous Wavelet Transform (CWT	No.	High-precision time–frequency positioning	Adjustable scale parameters adapt to different frequencies	[27,28]
	Discrete Wavelet Transform (DWT)	No.	Signal denoising for valve internal leakage	The db wavelet is optimal, and the entropy sensitivity is increased by 30%	[31,32]
	Wavelet Packet Decomposition (WPD)	No.	Complex frequency band feature extraction	Higher-frequency band resolution	[28,30]
	Empirical Mode Decomposition (EMD)	No.	Nonlinear signal processing	The modal aliasing is severe	[34,62]
	Local Mean Decomposition (LMD)	No.	Alleviate modal aliasing	The endpoint effect still exists	[34]
	Variational Mode Decomposition (VMD)	ISSA optimizes the number of layers/penalty factor	Diagnosis of valve faults in nuclear power plants	The average accuracy rate is high, superior to LMD and EMD	[34,35,36]
Nonlinear analysis	Multi-scale Entropy (MSE)	No.	Complexity assessment	Sensitive to noise	[38]
	Improved multi-scale entropy (LMMSE/EMSE)	No.	Bearing fault diagnosis	The feature recognition rate is 92% and the noise robustness is strong	[38]
	Time difference of arriva (TDOA)	No.	Location of cracks in liquid pipelines	The false alarm rate decreased from 35% in the traditional method to 8%	[63]
	Recursive Quantitative Analysis (RQA)	No.	Metal dislocation dynamics	The prediction error of plastic instability is less than 10%	[39]
Traditional machine learning	Support Vector Machine (SVM)	SSA optimizes nuclear parameters	Classification of internal leakage in valves	The accuracy rate of SSA-SVM is 99%	[41]
	K-nearest Neighbor (KNN)	No.	Simple leakage classification	The accuracy rate is 90–93%	[32,63,64]
	Random Forest (RF)	ISSA optimizes the feature subset and tree parameters	Pipeline leakage flow prediction	R^2^ = 0.894, MAE = 8.22 L/min	[49]
	BP neural network	SSA optimizes the initial weights	Prediction of internal leakage rate of ball valves	Error <6%	[50]
	AdaBoost.M1	Relief-F feature selection	Classification of internal leakage status of valves	Small-sample overfitting control	[48]
Deep learning	Deep Belief Network (DBN)	No.	Fault diagnosis of electric valves	The input effect of time-domain features is the best	[51]
	Convolutional Neural Network (CNN)	No.	Classification of pipeline leakage images	Accuracy rate: 95%	[52]
	CBAM-CNN	No.	Classification of damage to composite materials	Accuracy rate: 98.87%	[53]
	1D-CNN	Domain Adaptation (MDD/DANN)	Cross-model diagnosis of diesel engine exhaust valve faults	The accuracy rate of MDD is the best	[59]
	LSTM	No.	Leakage in the gas–liquid two-phase flow pipeline	Accuracy rate: 98.4%	[7]
	MFCC-LSTM	No.	Leakage detection in complex working conditions	The identification rate of crack leakage is 100%	[7]
	Variational Autoencoder (VAE)	No.	Small-sample leakage detection	The reconstruction error detection is abnormal	[56]
	Growth Neural Gas Network (GNG)	No.	Unlabeled pipeline leakage detection	The leakage sensitivity of 0.3mm is 96.5%	[57]
Optimization algorithm	Sparrow Search Algorithm (SSA)	Independent optimizer	General hyperparameter optimization	The convergence speed is 20% faster than that of PSO	[9,28,42,49]
	Improve SSA (ISSA)	Reverse learning + random walk	Complex nonlinear optimization	Avoid premature convergence and increase R^2^ by 5%	[34,49]
	Particle Swarm Optimization (PSO)	Optimize the parameters of SVM	Replace the comparison benchmark of SSA	Prone to fall into local optimum	[36]

## Data Availability

Data will be made available upon request.

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
