# Peer review of "Review of Acoustic Emission Detection Technology for Valve Internal Leakage: Mechanisms, Methods, Challenges, and Application Prospects"

_sensors, 2025, doi:10.3390/s25144487_

Round 1
Reviewer 1 Report
Comments and Suggestions for Authors
I think it is an interesting and large review concerning AE for valve internal leakage detection. I think Table 3-1 is good. On the other hand, the details of the AE experimental study are missing. It is a very long article. Perhaps it might be a good idea to publish two articles instead of one? Below are correction proposals:
I think only family names should be used in literature references, I can see first names both in reference table and in the text e.g. (l.246, l.263, l.434, l923, l.925, l.927 etc.).
I think conclusion is part of the results e.g. Table 5-1 (l.855).
Dots are missing e.g. in abstract (l.17)
Support vector machine (SVM) (l.524)
I think there should be no Chinese text in references (l.941)
Section 4. Experiment … (l.762)
You should report detailed information about the used AE-measurement system, signal analyses, and data handling. I think it would be important to have more details about AE measurements, AE-signals and parameters used in the model to have more solid results. I think you should show e.g. AE-signals and/or parameters from good valves and valves with defects, too. You should show AE-data used in the models e.g. used frequencies. How many inputs are fed in the models?
For me, the results seems to be too good e.g. coefficient of determination is 1 (l.830). It seems that there might be very similar data used both in training and testing. You should use more valves if you want to generalize the results. I think the models are working fine with the tested valves but not necessarily with other types of valves. The defects were self-made, which affect the results. Perhaps the models work only with certain types of defects similar to defects you made. If you have very old valve with wear fault, the response might be very different.
I think the linear regression prediction model is missing (l.851)
The text is wrong way around in Fig 4.5c, Fig. 4.7c (l.843, l.853)
Author Response
Comment 1:I think only family names should be used in literature references, I can see first names both in reference table and in the text e.g. (l.246, l.263, l.434, l923, l.925, l.927 etc.).
Response 1:I have changed all the citations to the form you requested.
Comment 2:I think conclusion is part of the results e.g. Table 5-1 (l.855).
Response 2:I have deleted the experimental section from the original manuscript, and the detailed experimental contents may be refined into a separate article.
Comment 3:Dots are missing e.g. in abstract (l.17)
Response 3:As a non-native English speaker, I didn't fully understand your comment. I have rewritten the abstract and would appreciate it if you could check whether there are any inappropriate parts.
Comment 4:Support vector machine (SVM) (l.524)
Response 4:I have written out the full name of SVM at l546.
Comment 5:I think there should be no Chinese text in references (l.941)
Response 5:I have completed the modification of all Chinese characters in the references.
Comment 6:Section 4. Experiment … (l.762)
You should report detailed information about the used AE-measurement system, signal analyses, and data handling. I think it would be important to have more details about AE measurements, AE-signals and parameters used in the model to have more solid results. I think you should show e.g. AE-signals and/or parameters from good valves and valves with defects, too. You should show AE-data used in the models e.g. used frequencies. How many inputs are fed in the models?
For me, the results seems to be too good e.g. coefficient of determination is 1 (l.830). It seems that there might be very similar data used both in training and testing. You should use more valves if you want to generalize the results. I think the models are working fine with the tested valves but not necessarily with other types of valves. The defects were self-made, which affect the results. Perhaps the models work only with certain types of defects similar to defects you made. If you have very old valve with wear fault, the response might be very different.
I think the linear regression prediction model is missing (l.851)
The text is wrong way around in Fig 4.5c, Fig. 4.7c (l.843, l.853)
Response 6:I have deleted the experimental section from the original manuscript, and the detailed experimental contents may be refined into a separate article.
Reviewer 2 Report
Comments and Suggestions for Authors
This paper presents a comprehensive review of recent advances in the application of acoustic emission (AE) technology for internal valve leakage detection. It covers theoretical foundations (e.g., Lighthill’s acoustic analogy), signal processing methods (parameter analysis, time-frequency analysis, nonlinear dynamics), intelligent algorithms (SVM, GPR, deep learning), and experimental validation. The research has clear engineering relevance, a well-structured layout, and a thorough literature review. However, the paper requires minor revisions to improve the clarity of theoretical derivations, citation consistency, and depth of discussion. Recommendation: Minor Revision
- Abstract Lacks Clear Highlight of Innovations, Revise the opening paragraph to clearly outline three main contributions.
- The formulation in the theoretical basis section lacks clarity in variable definitions and assumptions (e.g., the use of Curle’s source term in multiphase flow needs clearer context). Ensure all symbols are defined upon first use and check that all equations are dimensionally consistent and clearly linked to physical phenomena.
- Some figures and tables are referenced inconsistently. For example, the description of AE signal features in Section 4.2 does not match the annotations in Figure 3-2. Double-check that all figure captions and references in the text correspond to the correct content. Additionally, the comparison across Tables 4-1 to 4-5 (model performance metrics) would benefit from visualization:
- Add bar charts to illustrate RMSE or accuracy across models to aid intuitive comparison.
- The review currently emphasizes conventional ML techniques but underrepresents recent advances in deep learning.
- Include references from the last 3 years, especially those applying CNNs or transformers for AE signal classification, and discuss the trade-off between model accuracy and inference speed (e.g., GPR vs. deep learning for real-time systems). Add discussion on valve-type-specific AE characteristics: the current focus is on gate valves; include ball valves and globe valves and their AE response differences.
- Several grammatical and stylistic inconsistencies exist throughout the manuscript.
The English could be improved to more clearly express the research.
Author Response
Comment 1: Abstract Lacks Clear Highlight of Innovations, Revise the opening paragraph to clearly outline three main contributions.
Response 1: I have rewritten the entire abstract and would like to request your review.
Comment 2: The formulation in the theoretical basis section lacks clarity in variable definitions and assumptions (e.g., the use of Curle’s source term in multiphase flow needs clearer context). Ensure all symbols are defined upon first use and check that all equations are dimensionally consistent and clearly linked to physical phenomena.
Response 2: I added background explanations from lines 172 to 175 on Page 5 and supplemented the previously missing description of physical quantity V from lines 178 to 180 on Page 5.
Comment 3: Some figures and tables are referenced inconsistently. For example, the description of AE signal features in Section 4.2 does not match the annotations in Figure 3-2. Double-check that all figure captions and references in the text correspond to the correct content. Additionally, the comparison across Tables 4-1 to 4-5 (model performance metrics) would benefit from visualization:Add bar charts to illustrate RMSE or accuracy across models to aid intuitive comparison.
Response 3: In response to suggestions from other reviewers, I have removed the experimental section from the original manuscript. The detailed experimental contents can be refined into a separate article.
Comment 4: The review currently emphasizes conventional ML techniques but underrepresents recent advances in deep learning.Include references from the last 3 years, especially those applying CNNs or transformers for AE signal classification, and discuss the trade-off between model accuracy and inference speed (e.g., GPR vs. deep learning for real-time systems). Add discussion on valve-type-specific AE characteristics: the current focus is on gate valves; include ball valves and globe valves and their AE response differences.
Response 4: I inserted a descriptive paragraph about Convolutional Neural Network (CNN) from l 710 to l 720 on Page 20 and added a supplementary section on CNN from l 743 to l 753 on Page 21. Two relevant references have also been incorporated. Regarding the discussion on the AE characteristics of different valves, I believe your opinion is reasonable. However, this paper tends to focus on providing acoustic emission internal leakage detection methods, and it is difficult to have a unified standard for measuring the differences in signal characteristic responses of different valves. The specifications and flow rates of valves used in different experiments vary significantly, making it challenging to compare their AE characteristics (it is unclear whether the differences in signal characteristics are due to valve structure, flow rate, leakage mode, or other factors). Therefore, I think adding this content is slightly beyond the scope of this paper. (This is just my thought and discussion. If you have other ideas or opinions, please feel free to let me know, and I would be grateful.)
Comment 5: There are some grammatical and stylistic inconsistencies throughout the manuscript.
Response 5: I have revised the grammatical and lexical issues that I could identify, and unified all citations in the text to the format of "Familyname et al. [1]proposed".
Reviewer 3 Report
Comments and Suggestions for Authors
The paper presents an overview of acoustic emission methods for valve leak detection and monitoring. Machine learning models such as SVM, neural networks, SSA, and GPR are described, which improve the accuracy and adaptability of leak detection. The text of the manuscript is well structured, but there are a number of significant comments.
- The text contains figures and tables but there are no references to them in the text.
- The numbering of figures and tables should be continuous throughout the text.
- Description of the figures should be given in the text. For example Figure 3.4 (page 18), What does this framework means? Each Figure given in the text must be explained.
- Please check the style of reference in the text. The common style is "Ahmad Braydi et al. [21] presented..." (Page 9).
- Page 6, Table 2-1. The sentence "In the previous text, (1.1)" is unclear. There is no equation (1.1) or (1.2). It is better to write the equations in a table or delete them.
- Page 13, line 439-447. Symbols introduced by Igor Rastegaev et al. [37] should be given only with the corresponding Figure. Please improve the text.
- There are no descriptions of the Table 4.1- Table 4.5 and corresponding Figures 4.3 - 4.7. Section 5 should be combined with the results of the models. It is better to add a short description for each model and summarize them in Table 5.
Author Response
Comments 1: The text contains figures and tables but there are no references to them in the text.
Response 1: I have rechecked my article and the images, and added descriptions to all images that were previously undescribed.
Comments 2: The numbering of figures and tables should be continuous throughout the text.
Response 2: I have checked the images and tables throughout the article and renumbered all images in sequential order.
Comments 3: Description of the figures should be given in the text. For example Figure 3.4 (page 18), What does this framework means? Each Figure given in the text must be explained.
Response 3: I added a paragraph explaining the content of Figure 11 under Figure 11 on page 19 from l 685 to l 694. Additionally, I inserted a paragraph describing the content of Figure 6 under Figure 6 on page 7 from l 207 to l 214.
Comments 4: Please check the style of reference in the text. The common style is "Ahmad Braydi et al. [21] presented..." (Page 9).
Response 4: I have revised all parts of the article that did not conform to the formatting requirements. Since there were too many revision points, I will not specify each modification location here.
Comments 5: Page 6, Table 2-1. The sentence "In the previous text, (1.1)" is unclear. There is no equation (1.1) or (1.2). It is better to write the equations in a table or delete them.
Response 5: I have changed the formula numbers in the table on page 5 to (1) and (2) respectively. These two formulas are too long to be placed in the table, which affects the aesthetics.
Comments 6: Page 13, line 439-447. Symbols introduced by Igor Rastegaev et al. [37] should be given only with the corresponding Figure. Please improve the text.
Response 6:I have changed the symbolic expressions from l 456 to l 458 on page 13 into text descriptions (the original symbolic images are not aesthetically pleasing, so they are not placed here).
Comments 6: There are no descriptions of the Table 4.1- Table 4.5 and corresponding Figures 4.3 - 4.7. Section 5 should be combined with the results of the models. It is better to add a short description for each model and summarize them in Table 5.
Response 6:In response to suggestions from other reviewers, I have removed the experimental section from the original manuscript. The detailed experimental contents may be elaborated into a separate article.
Reviewer 4 Report
Comments and Suggestions for Authors
In my opinion, most of the numbers (values) given in the paper are for too many valid points. I miss the determination of the accuracy of individual calculations and estimates. For example, Table 5.1 RMSE = 0.54411 => I expected 0.544 p.m. 0.001.
Where can I find data and programs to test the functionality of the systems?
It is not clear to me how the noise was eliminated during the measurement.
It is not clear how the AE was recorded and how it was analysed.
How did the authors test analytical models - machune learning, etc.?
Comments on the Quality of English Language
I don't want to argue because I am not a native or educated English speaker. I understand the text.
Author Response
Comments 1 to 5:
In my opinion, most of the numbers (values) given in the paper are for too many valid points. I miss the determination of the accuracy of individual calculations and estimates. For example, Table 5.1 RMSE = 0.54411 => I expected 0.544 p.m. 0.001.
Where can I find data and programs to test the functionality of the systems?
It is not clear to me how the noise was eliminated during the measurement.
It is not clear how the AE was recorded and how it was analysed.
How did the authors test analytical models - machune learning, etc.?
Response 1 to 5: In response to suggestions from other reviewers, I have deleted the experimental section from the original manuscript. The detailed experimental contents may be refined into a separate article
Round 2
Reviewer 1 Report
Comments and Suggestions for Authors
I think it is an interesting and large review concerning AE for valve internal leakage detection. I think Table 3-1 is good. I think it a good review.
I found next minor issues:
l131-l134: I think you are talking about the results which you removed and they are not shown; if you like to add this, you should first publish your experimental results using GPR
l210: Fig 1.1. should be Fig. 1
l855: vibration -> Vibration
In References: only last-names, and initials should be used e.g. l837, l839, l841, l854 etc.
Author Response
I have made detailed revisions to your suggestions. I have deleted the paragraphs you pointed out and made corresponding changes to the rest. Since the content is simple, I won't list them all.
Reviewer 3 Report
Comments and Suggestions for Authors
Correction accepted
Author Response
Thank you for your evaluation.